∂ | **Open Peer Review** | Clinical Microbiology | Research Article

# CAMP-negative *Streptococcus agalactiae* strains exhibited complete or partial chromosomal deletions of the CAMP-factor encoding gene *cfb*

Xixi Lai,[1] Meihong Chen,[1] Jianwei Wang,[1] Junjun Wang,[1] Hui Lv,[1] Haihua Xie,[1] Wenjuan He,[1] Dongjie Chen,[1] Yi Huang,[1,2,3,4] Pengwei Cai,[1] Lilan Zheng[1]

**ABSTRACT** Universal antepartum group B streptococcus (GBS) screening and intrapartum antibiotic prophylaxis (IAP) have effectively reduced early-onset GBS infections. However, GBS strains with chromosomal deletions affecting the *cfb* gene may produce false negatives in both the CAMP test and *cfb*-based molecular diagnostics, potentially increasing the risk of neonatal infections. Vaginal swabs were collected from pregnant women at 35–37 weeks of gestation in our hospital and cultured on agar. Suspected GBS strains were initially identified using the CAMP test and then confirmed with the VITEK-2 system. CAMP-negative GBS strains underwent additional testing by qPCR, 16S rDNA, serotyping, and multilocus sequence typing (MLST). PCR for the *cfb* gene and whole-genome sequencing were performed on CAMP-negative strains. From 5,794 samples, 526 (9.1%) GBS strains, including 19 (3.6%) CAMP-negative strains and 2 strains from the same patient, were isolated. All 19 CAMP-negative strains were serotypes III and ST862. Among these strains, only one strain was *cfb* positive by qPCR, whereas all tested positive with a multitarget qPCR kit for *cfb* and *cps*. PCR amplification upstream of the *cfb* gene produced a specific band in strain PP669713 only, suggesting N-terminal *cfb* gene retention in PP669713 and complete *cfb* loss in the other strains. Whole-genome sequencing confirmed a chromosomal deletion in PP669713. Antibiotic susceptibility testing revealed no resistance to penicillin. However, CAMP-positive strains presented a greater prevalence of resistance to ciprofloxacin, and levofloxacin than CAMP-negative strains did. Our study highlights the potential risk of missed GBS detection using CAMP tests and *cfb*-targeted molecular assays.

**IMPORTANCE** Our work makes several novel contributions to the field. (i) We report the first documented case of a C-terminal deletion of the *cfb* gene in a CAMP-negative GBS strain, demonstrating that both N-terminal and C-terminal regions are essential for cohemolytic activity. (ii) Our findings reveal that CAMP-negative GBS strains (3.6% of isolates) are more prevalent than previously recognized, with most cases resulting from complete chromosomal deletions of the *cfb* gene. (iii) We provide evidence that single-target molecular assays targeting only the *cfb* gene may miss GBS detection, highlighting the necessity for multi-target approaches in clinical diagnostics. (iv) We demonstrate a unique antibiotic resistance pattern in CAMP-negative strains, showing significantly lower resistance to certain antibiotics compared to CAMP-positive strains.

**KEYWORDS** *Streptococcus agalactiae*, CAMP-negative, *cfb*, molecular test, multitarget

Address correspondence to Lilan Zheng, 19211010055@fudan.edu.cn, Pengwei Cai, 916914369@qq.com, or Yi Huang, hyi8070@126.com.

Xixi Lai and Meihong Chen contributed equally to this article. Author order was determined in order of increasing seniority.

The authors declare no conflict of interest.

See the funding table on p. 13.

*S*treptococcus agalactiae (group B streptococcus, GBS) primarily colonizes the maternal gastrointestinal and urogenital tracts. It is a significant perinatal pathogen associated with approximately 150,000 stillbirths or infant deaths each year (1–3). Before the

implementation of intrapartum antibiotic prophylaxis (IAP), nearly half of GBS-positive pregnant women transmitted the bacteria to their newborns. Approximately 1%–2% of these infants develop invasive diseases, such as neonatal sepsis, meningitis, or pneumonia (4, 5). In the United States, the introduction of IAP and consensus guidelines for GBS screening in 1996, which were updated in 2010 and 2019, led to a decline in neonatal infections from 1.8 cases per 1,000 live births in 1990 to 0.23 cases per 1,000 live births by 2015 (6–11).

In the context of group B streptococcus (GBS) prevention strategies, two primary approaches are employed globally: universal screening-based prophylaxis (SBP) and risk-based prophylaxis (RBP). SBP involves screening all pregnant women for GBS, regardless of their risk factors, typically through vaginal swab cultures collected at 35–37 weeks of gestation. This approach, considered universal screening, aims to identify and treat all GBS-colonized women with antibiotics during labor to prevent early-onset GBS disease in their infants. On the other hand, RBP targets specific pregnant women who have identified risk factors for GBS colonization or disease, such as a previous history of GBS colonization or infection, preterm delivery, or the presence of certain maternal conditions like diabetes or gestational hypertension. In such programs, only women with these risk factors undergo GBS screening and, if positive, receive antibiotic prophylaxis.

A 2017 systematic review covering 95 countries revealed that 60 countries adopted an IAP policy. Among these, 35 countries, including Belgium, Finland, and Spain, utilized SBP, whereas 25 countries, such as the Netherlands, New Zealand, and the United Kingdom, opted for RBP (12). In China, some regions recommend universal screening for GBS in pregnant women, whereas others suggest a risk factor-based approach due to varying resources (13, 14).

Although universal screening can reduce early-onset GBS disease (EOGBS), it has limitations. These include high healthcare costs associated with over-detection and the risk of antimicrobial resistance or microbiota dysregulation in mothers and infants resulting from excessive antibiotic use, particularly in low-incidence areas (13, 15–17). Furthermore, a risk factor-based strategy can also maintain EOGBS incidence at relatively low levels (18). Therefore, selecting the most suitable approach requires careful consideration of regional epidemiological data, the costs of screening, and other relevant factors. Additionally, developing an effective maternal GBS vaccine represents the best method for preventing EOGBS. Continued surveillance of GBS serotype switching is also crucial for vaccine research and development (19).

Presumptive GBS detected on agar media can be identified using several phenotypic methods. Among these, the CAMP test, named after its discoverers Christie, Atkins, and Munch-Peterson, has long been a reliable diagnostic tool in clinical microbiology laboratories since its introduction in 1944 (20, 21). The CAMP reaction produces an arrow-shaped enhanced hemolytic zone on a sheep blood agar when GBS is cultured alongside *Staphylococcus aureus* (22, 23). Given that CAMP is present in nearly all isolated human GBS strains, the CAMP test is widely used for GBS identification in diagnostic laboratories. Recently, nucleic acid amplification testing (NAAT) has been employed, with the *cfb* gene encoding the CAMP factor serving as a common target gene (21, 24, 25). Compared with culture-based methods, this method is more rapid and sensitive, significantly reducing the turnaround time (TAT) and lowering the false negative rate (24, 26).

However, GBS may evade detection if the diagnostic methodology relies solely on the CAMP reaction, especially in cases involving phenotypically and partly genotypically CAMP factor-deficient strains (27). Although CAMP-negative GBS was first reported in bovine mastitis in the early 21st century, the isolation of two CAMP-negative GBS strains from pregnant women in 2016 attracted significant research interest (28–30). Some CAMP-negative strains possess normal-sized *cfb* genes but cannot express the CAMP factor or have low synthesis and activity of the CAMP factor (27, 29). Fortunately, these GBS strains can still be detected via PCR. However, NAATs that depend on the *cfb* gene may fail to identify strains if the *cfb* gene is modified or absent (31–33). Studies indicate

that *cfb* gene-deficient GBS strains account for approximately 1%–7.9% of strains across various regions, underscoring the need for further research into the frequency and mechanisms of CAMP-negative GBS strains from different geographical regions (33, 34).

This study isolated 19 CAMP-negative strains and analyzed their molecular characteristics and prevalence of antibiotic resistance. Our results showed a CAMP-negative rate of 3.6%, which is lower than the level reported by Zhou et al., but significantly higher than the 1% previously reported in the literature. Additionally, our study highlights the potential risk of false-negative diagnoses when relying solely on CAMP tests or PCR assays targeting the *cfb* gene. Furthermore, our study revealed various chromosomal deletions of the *cfb* gene in CAMP-negative GBS strains.

## MATERIALS AND METHODS

### GBS isolation

The study involved 5,794 vaginal samples collected during routine GBS antenatal screening of pregnant women at 35–37 weeks of gestation. These samples were taken between June 2019 and December 2020 at Fujian Provincial Hospital. A total of 526 strains (9.1%) of *S. agalactiae* were isolated. Among these strains, 19 strains (3.6%) were CAMP negative, and two strains were simultaneously isolated from the same participant.

Vaginal swabs were promptly placed into ESwab transport medium (ESwab, Copan Diagnostics, Brescia, Italy) and subcultured onto GBS chromogenic agar plates (Autobio Diagnostics, Zhengzhou, China) within 2 h. The agar plates were incubated at 37°C for 24––48 h. The purple colonies observed on the chromogenic agar were selected and cultivated on 5% sheep blood agar (Autobio Diagnostics, Zhengzhou, China) for an additional 24–48 h, followed by identification through subsequent processes.

### Identification of GBS

The identification of group B streptococcus (GBS) was conducted using four methods: the VITEK-2 automatic identification system (BioMérieux, France), Matrix-Assisted Laser Desorption/Ionization Time-of-Flight Mass Spectrometry (MALDI-TOF MS, Autobio Diagnostics, Zhengzhou, China), real-time fluorescence quantitative PCR (qPCR), and 16S rDNA gene sequencing. All suspected GBS strains were initially identified using the VITEK-2 system. CAMP-negative GBS strains were further confirmed through MALDI-TOF MS, qPCR, and 16S rDNA gene sequencing.

For qPCR, two different GBS nucleic acid detection kits were used. The first kit, manufactured by TianLong Technology (Suzhou, China), targeted the *cfb* gene. The primer set sequences for this kit included a forward primer (F1) 5′-ATGATGTATCTAT CTGGAACTCTAGTG-3′ and a reverse primer (R1) 5′-CGCAATGAAGTCTTTAATTTTTC-3′. Alternative primers were also available, with F1 5′-AGC GAA TTA TAC CGT TCA GG-3′ and R1 5′-GTC TCT TCC TCG TAT TGC TG-3′. The qPCRswere performed on an ABI 7500 Real-Time PCR System (Applied Biosystems, USA) with thermocycler parameters consisting of an initial denaturation at 95°C for 3 minutes, followed by 40 cycles of 94°C for 15 seconds (denaturation) and 60°C for 30 seconds (annealing/extension). Fluorescence detection was achieved using SYBR Green I intercalating dye. Post-amplification melting curve analysis was conducted from 60°C to 95°C with a heating rate of 0.5°C per second. A single sharp peak at approximately 82°C was considered acceptable, indicating specific amplification of the target *cfb* gene.

The second qPCR kit, manufactured by Sansure Biotech, Inc. (Changsha, China), targeted both the cfb and cps genes. For the *cfb* gene, the primer sequences were F1 5′-AGC GAA TTA TAC CGT TCA GG-3′ and R1 5′-GTC TCT TCC TCG TAT TGC TG-3′, along with a probe 5′-FAM-TGG TCA GCA TGT TGA TGA TGA TGA TGA-BHQ1-3′. For the cps gene, the primer sequences were F1 5′-GCA TAA TGG TCA TCA TCA TCA TCA-3′ and R1 5′-TCA GCA TGT TGA TGA TGA TGA TGA-3′, along with a probe 5′-HEX-CGA TGA TGA TGA TGA TGA TGA TGA TGA-BHQ1-3′. The qPCRs for this kit were also performed on an

ABI 7500 Real-Time PCR System (Applied Biosystems, USA) with thermocycler parameters consisting of an initial denaturation at 94°C for 5 minutes, followed by 40 cycles of 94°C for 15 seconds (denaturation) and 57°C for 30 seconds (annealing/extension). Fluorescence detection was achieved using dual-labeled probes (FAM and HEX) with BHQ1 quenchers.

DNA extraction and 16S rDNA gene sequencing were performed by RuiBiotech (Beijing, China). Several bacterial colonies were picked into sterile tubes, stored on dry ice, and sent to the company. The 16S region was amplified using universal primers (27F: 5′-AGAGTTTGATCCTGGCTCAG-3′; 1492R: 5′-GGTTACCTTGTTACGACTT-3′). The resulting 16S rDNA sequence data were analyzed using the GenBank database at NCBI. For species-level identification, a sequence match of ≥99% to *Streptococcus agalactiae* in the NCBI database was required.

*S. agalactiae* (ATCC13813) served as a positive quality control, whereas *Enterococcus faecalis* (ATCC29212) was used as a negative quality control in these assays.

## CAMP test

Presumptive GBS isolates were cultured on 5% sheep blood agar (Autobio Diagnostics, Zhengzhou, China) at 37°C in a 5% $CO_2$ environment for 24 h. They were placed adjacent to *Staphylococcus aureus* (ATCC25923) to stimulate the production of β-hemolysin. A positive reaction was indicated by the formation of a characteristic arrow-shaped hemolysis surrounding GBS, whereas its absence signified a negative result (9). *S. agalactiae* (ATCC13813) was used as a positive control, and *E. faecalis* (ATCC29212) served as a negative control.

## DNA extraction

Total bacterial DNA was prepared via the boiling method (35). Several colonies were picked into 200 µL of DNase-free water using an inoculation loop. The mixture was vortexed for 30 s and incubated at 100°C for 10 min. The boiling product was then centrifuged at 12,000 $g \cdot min^{-1}$ for 5 min, and the supernatant was stored at −80°C.

## Typing methods

GBS typing was performed using two methods: capsular serotyping and sequence typing (ST). Serotyping was assessed via multiplex PCR, whereas ST was conducted via multilocus sequence typing (MLST) as described in previous studies (36, 37).

## Antibiotic susceptibility testing

Antibiotic susceptibility testing was performed with the VITEK-2 system with an AST-GP67 card (BioMérieux, France) following the manufacturer's instructions. *Staphylococcus aureus* (ATCC29213) and *E. faecalis* (ATCC29212) served as quality control strains to verify the accuracy of the results.

## Amplification and sequencing of *the cfb* gene

Bacterial DNA was extracted as described earlier. The *cfb* gene was amplified using two pairs of primers targeting the upstream and downstream regions of the *cfb* gene, as previously reported (33). These primers were designated F1, R1 and F2, R2 (F1: 5′-ATGATG TATCTATCTGGAACTCTAGTG-3′; R1: 5′-CGCAATGAAGTCTTTAATTTTTC-3′; F2: 5′ -TGGTAGTC GTGTAGAAGCCTTA-3′; R2: 5′-TCCAACAGCATGTGTGATTGC-3′). To avoid missed detection due to modifications or deficiencies in the *cfb* gene, an additional set of primers, F3 and R3, was designed to amplify the entire *cfb* gene region (F3: 5′-GGGGAAAAGAAAG CGCTTGACG-3′; R3: 5′-GGTGACATCGTTCTACTATTATGAC-3′). The specific target sites and amplified fragments are shown in Fig. 2A. All amplified fragments were analyzed by agarose gel electrophoresis, and only fragment A was sequenced by RuiBiotech (Beijing, China). For the agarose gel electrophoresis, a 2% agarose gel was prepared and used. The

electrophoresis was performed using a Bio-Rad PowerPac Basic electrophoresis power supply, with a voltage set to 100 V and a run time of approximately 45 minutes. The electrophoresis equipment used was a Bio-Rad Gel Doc XR + System. The detection method involved staining the gel with ethidium bromide, a fluorescent dye that binds to DNA, and visualizing the bands under UV light using a digital camera integrated into the Gel Doc XR + System. These parameters ensured clear and distinct amplification of the targeted *cfb* gene fragments, as shown in Fig. 2B through E.

## Whole-genome sequencing

Whole-genome sequencing and analysis were performed by RuiBiotech (Beijing, China) following the standard protocol from Illumina. Sequencing libraries were prepared from extracted genomic DNA using the Watchmaker DNA Library Prep Kit (Watchmaker Genomics, 7K0019-096) and sequenced on an Illumina HiSeq sequencer. The filtered reads were assembled using Megahit software and aligned to the complete GBS reference genome of the same serotype via Burrows–Wheeler Aligner (BWA) software. Only strain PP669713 was subjected to whole-genome sequencing, with the complete genome of NGBS128 (accession number NZ_CP012480) serving as the reference for alignment of strain PP669713.

## Statistical analysis

The data were analyzed using a combination of statistical methods to ensure robust and comprehensive results. For categorical variables, we employed Fisher's Exact Test to compare proportions between CAMP-positive and CAMP-negative isolates, particularly for small sample sizes. To address the imbalance in sample sizes (CAMP-positive: $n = 490$; CAMP-negative: $n = 19$), we also used Bootstrap Resampling, which involved repeated random sampling with replacement from the larger group to generate data sets of equal size to the smaller group. This method provided more reliable *P*-values and accounted for variability in the data.

To quantify the relative difference in antibiotic resistance between the two groups, we calculated Relative Risk (RR) along with 95% Confidence Intervals (*CIs*). An RR > 1 indicates a higher risk of resistance in CAMP-positive isolates, while an RR < 1 suggests a lower risk. The 95% CI provides a range of plausible values for the RR, allowing for a more nuanced interpretation of the results.

Additionally, we conducted a Bayesian Posterior Probability analysis to estimate the probability that the resistance rate in CAMP-positive isolates is higher than in CAMP-negative isolates. This approach incorporates prior knowledge and provides a more interpretable measure of the difference between the two groups.

For continuous variables, we compared means using two-sample *t*-tests. For categorical variables, we utilized both the Pearson $\chi^2$ test and Fisher's exact test, depending on the expected cell counts. All statistical tests were two-tailed, with a significance threshold set at $P < 0.05$.

All analyses were performed using SPSS version 20.0 (IBM Corp., Armonk, NY, USA).

## RESULTS

### Isolation and characterization of 19 CAMP-negative GBS strains

Between June 2019 and December 2020, we isolated 526 strains of *S. agalactiae* from 5,794 vaginal samples collected from pregnant women at 35–37 weeks of gestation at our hospital. Two strains were isolated from the same participant during the same session. One isolate appeared as a smooth grayish-white colony, whereas the other was slightly whiter. The GBS colonization rate during antenatal screening was 9.1%. Among the isolates, 19 were CAMP-negative but formed purple colonies on chromogenic agar plates (Fig. 1A and B). These plates enable quicker and easier GBS detection on the basis of pigment production. As depicted in Fig. 1B, all 19 isolates formed

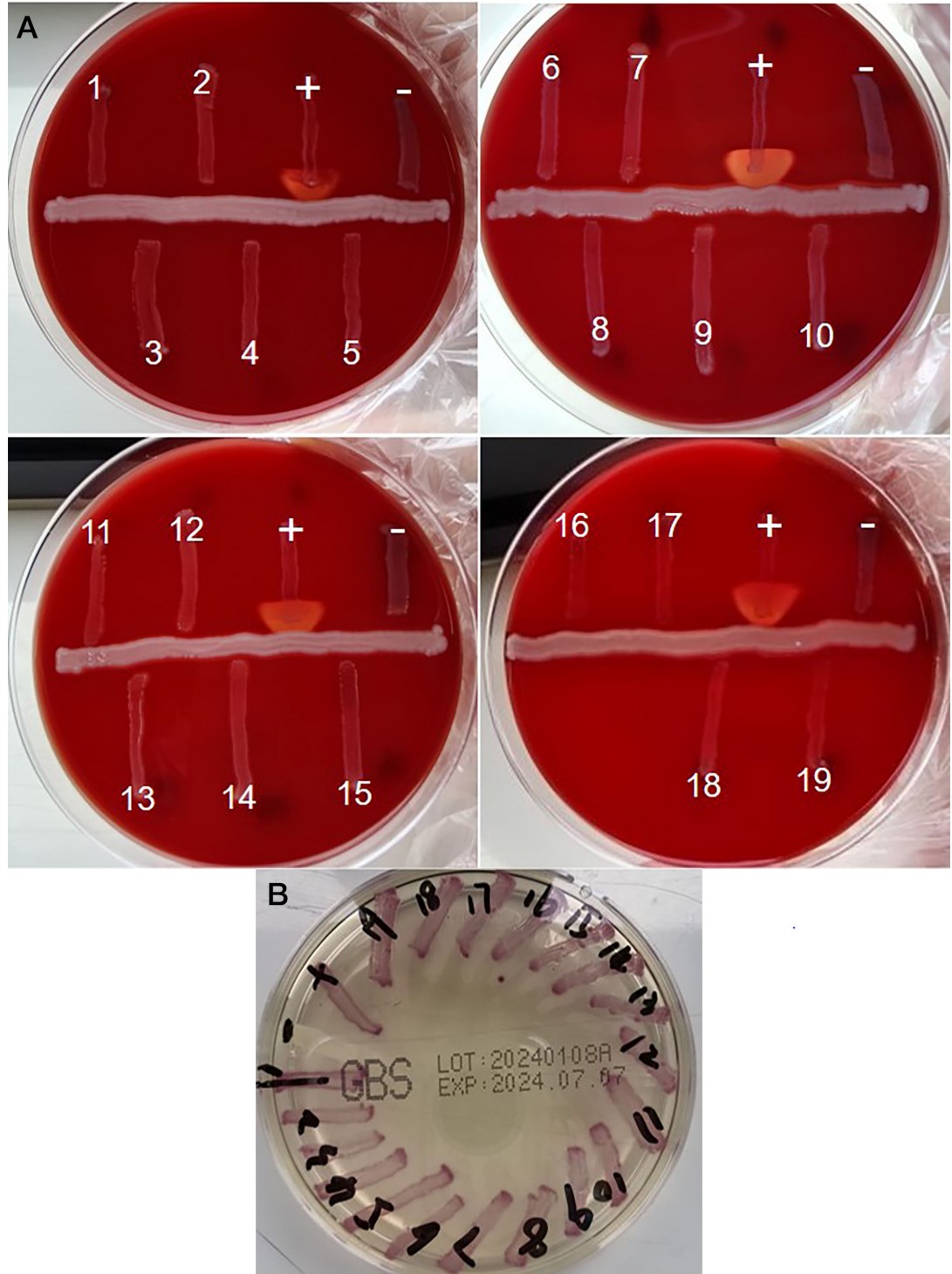

**FIG 1** (A) CAMP test for 19 suspected CAMP-negative *Streptococcus agalactiae* isolates. (A) Production of CAMP by GBS causes arrow-shaped hemolysis in the presence of Staphylococcus aureus on sheep blood agar. +, Positive control (*Streptococcus agalactiae*), −, negative control (*Enterococcus faecalis*). (B) Growth of 19 CAMP-negative GBS strains on chromogenic agar plates (B). GBS metabolizes chromogenic substances resulting in purple pigmented colonies on the chromogenic agar. +, Positive control (*Streptococcus agalactiae*), −, negative control (*Enterococcus faecalis*). The chromogenic agar exhibits a translucent light yellow color.

purple colonies. Specifically, when GBS strains grow on media containing these specific enzymatic substrates, the enzymes produced by the strains catalyze the breakdown

of the substrates, releasing chromogenic compounds. These chromogenic compounds then react with other components in the media, causing a purple color change around the colonies or throughout the entire medium. This color change is characteristic of GBS strains, enabling rapid and visual differentiation of GBS from other bacteria, thereby enhancing the efficiency and accuracy of identification. We further identified these presumptive GBS strains using the VITEK-2 and MALDI-TOF MS systems and revealed that CAMP-negative GBS strains comprised 3.6% of the samples. The clinical characteristics of the pregnant women from whom GBS was isolated can be found in Table 1. No significant differences were noted in the clinical characteristics of women with CAMP-positive vs CAMP-negative GBS strains, except with respect to pregestational BMI status.

To confirm the identification of GBS, we used two different qPCR kits: one targeting the *cfb* gene and another examining both the *cfb* and cps genes. In the single-target qPCR test, only PP669713 tested positive, whereas all other strains tested negative. Conversely, the results of the multitarget qPCR kit revealed that all 19 isolated strains were positive. The discrepancy observed between the two qPCR assays may be due to the presence of chromosomal deletions or mutations within the *cfb* gene in CAMP-negative GBS strains. While the single-target qPCR assay detected the presence of the *cfb* gene, the multitarget qPCR assay, which also targets the cps gene, provided additional confirmation. The qualitative and quantitative differences in the detection of the *cfb* gene between the two assays suggest potential genetic variations. Additionally, all CAMP-negative isolates were confirmed as GBS through 16S rDNA sequence analysis, with the data uploaded to NCBI GenBank. Serotyping and MLST analysis revealed that all CAMP-negative isolates were serotype III and ST-862, with the exception of strain PP669702.

TABLE 1 Clinical characteristics of pregnant women with GBS isolation[b]

| Group | Total | CAMP-positive | CAMP-negative | P |
|---|---|---|---|---|
| All patients, n (%) | 489 | 471 | 18[a] | |
| Age, median (IQR) | 30.11, (27.50, 32.00) | 30.09 (28.00, 32.00) | 30.56 (27, 33.5) | 0.09 |
| Elderly (age > 35), n (%) | 41 (8.4%) | 39 (8.3%) | 2 (11.1%) | 0.66 |
| Race and ethnicity | Asian | | | |
| Pregestational BMI status, n (%) | | | | **0.00** |
| Normal | 452 (92.4) | 446 (94.7%) | 6 (33.3) | |
| (18.5 ≤ BMI < 24.0) | | | | |
| Overweight | 23 (4.7) | 12 (2.5) | 11 (61.1) | |
| (24.0 ≤ BMI < 28.0) | | | | |
| Obesity | 14 (2.9) | 13 (2.8) | 1 (5.6) | |
| (BM ≥ 28.0) | | | | |
| Maternal parity, n (%) | | | | 0.05 |
| 1 | 184 (37.6) | 173 (36.7) | 11 (61.1) | |
| ≥2 | 305 (62.4) | 298 (63.3) | 7 (38.9) | |
| Comorbidity, n (%) | | | | |
| Gestational diabetes mellitus | 71 (14.5) | 68 (14.4) | 3 (16.7) | 0.74 |
| Gestational hypertension | 10 (2.0) | 8 (1.7) | 2 (11.1) | 0.05 |
| Thyroid disease | 57 (11.7) | 55 (11.7) | 2 (11.1) | 1.00 |
| Uterine fibroids | 34 (7.0) | 33 (7.0) | 1 (5.6) | 1.00 |
| Outcome of birth | | | | |
| PROM | 134 (27.4) | 131 (27.8) | 3 (16.7) | 0.42 |
| Preterm | 32 (6.5) | 32 (6.8) | 0 (0.0) | 0.62 |
| GBS disease | 7 (1.4) | 6 (1.3) | 1 (5.6) | 0.232 |

[a]Two CAMP-negative GBS strains were isolated from the same participant.
[b]PROM, premature rupture of membranes. IQR, inter quartile range. P values were calculated by Pearson $\chi$2 test or Fisher's exact test or one-way ANOVA test.

## CAMP-negative GBS strains presented complete or partial chromosomal deletions of the CAMP-factor encoding gene (*cfb*)

Previous studies have suggested that a negative CAMP test may result from abnormal expression or deficiency of the CAMP factor-encoding gene *cfb*. To investigate the mechanism behind the negative CAMP test in our study, we employed two pairs of primers targeting different regions of the *cfb* gene (fragments A and B) and designed an additional pair to target approximately 100 bp upstream and downstream of the *cfb* gene (fragment C). The results from agarose gel electrophoresis (Fig. 2B through D) indicated that, except for sample 13, no positive bands were detected in the other samples, suggesting a possible loss of the entire *cfb* gene in these strains. In sample 13, bands representing fragments A (260 bp) and D (390 bp) were present (Fig. 2B and E), but no positive bands were detected for fragments B and C. This implies that the negative CAMP test for strain PP669713 may be due to a C-terminal deletion of the *cfb* gene rather than gene downregulation.

To confirm that strain PP669713 lost a portion of the chromosome containing the *cfb* gene, we conducted whole-genome sequencing of the strain. The analysis (Fig. 3) revealed a 16,795 bp chromosomal deletion downstream of the *cfb* gene in GBS strain PP669713. This finding explains why the primers targeting this region failed to amplify any bands in previous experiments. These results conclusively demonstrate that the GBS strain PP669713 indeed has a chromosomal deletion at the 3′ end of the *cfb* gene.

### Antibiotic susceptibility

The antibiotic susceptibility results are presented in Table 2. As shown in the table, none of the GBS strains exhibited resistance to penicillin, ampicillin, or linezolid. The data in Table 2 also underscore notable disparities in antibiotic resistance between CAMP-positive and CAMP-negative isolates of group B streptococcus (GBS), particularly in relation to quinupristin, ciprofloxacin, and levofloxacin.

### DISCUSSION

In this study, 526 GBS strains were isolated from 5,794 vaginal swabs, resulting in a colonization rate of 9.1% among pregnant women. This finding aligns with previous reports (38). These results highlight the importance of antenatal GBS screening. Among the isolates, 19 strains were identified as CAMP-negative. Further analysis indicated that the CAMP-negative phenotype in all 19 isolates was due to a deficiency of the *cfb* gene, with a deficiency rate of 3.6%. This rate is lower than that reported in Shenzhen, China, but higher than that reported in other regions (32, 33). However, this discrepancy may be due to differences in sample selection, geographical variations, or variations in detection methods. Further research is needed to confirm this difference. In our study, due to differences in routine laboratory procedures, the traditional manual streaking culture method was used without enrichment processing, while this approach facilitated quicker results, which may also be a reason for the lower positive rate. To ensure the rigor of the study, we will strictly follow the guidelines for sample enrichment in subsequent research to improve the accuracy of detection.

CAMP, a pore-forming protein of GBS, causes arrow-shaped hemolysis in the presence of *Staphylococcus aureus* on sheep blood agar, known as the CAMP reaction. Since CAMP is found in nearly all GBS strains, the CAMP test has become a widely used method for identifying GBS. Moreover, the gene encoding the CAMP factor *cfb* is commonly targeted in NAATs. However, the emergence of CAMP-negative GBS strains since 2016 has challenged the previously held belief regarding their rarity. This emergence raises concerns about the potential limitations of molecular assays that rely exclusively on detecting the *cfb* gene for GBS identification.

Currently, two primary types of chromogenic media are extensively utilized for the detection and identification of group B streptococcus (GBS). The first type is Media Based on β-Hemolysis Characteristics, which capitalizes on the β-hemolytic properties of GBS

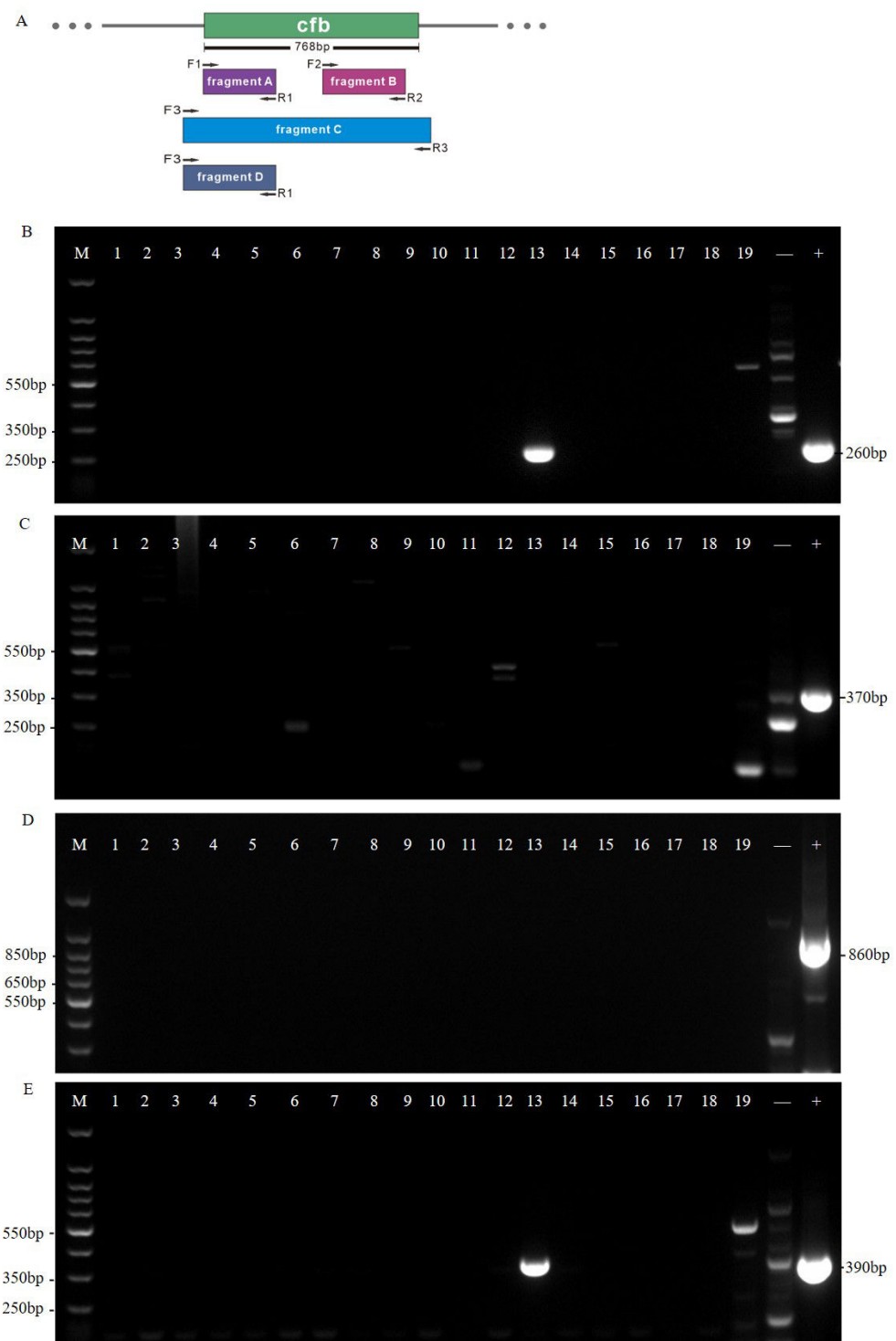

**FIG 2** PCR amplification results of the *cfb* gene in 19 isolated CAMP-negative GBS strains. (A) Primer design for *cfb* gene detection and targeting different regions of the gene. Fragment A is 260 bp, fragment B is 370 bp, fragment C is 860 bp, and fragment D is 390 bp. (B–E) Show the amplification results for fragments A–D as described in Fig. 2A, respectively. (B) Only sample 13 shows a single band (260 bp) serving as the positive control. (C) None of the samples show a band of the same size as the positive control. (D) None of the samples show a band of the same size as the positive control. (E) Only sample 13 shows a single band (390 bp) serving as the positive control. Sample 13 represents strain PP669713. 250 bp, 350 bp, 850 bp, and 350 bp ladders serve as size markers for Figures B–E, respectively. M: DNA Marker J (150–1,500 bp), +, positive control (*Streptococcus agalactiae*), −, negative control (*Enterococcus faecalis*).

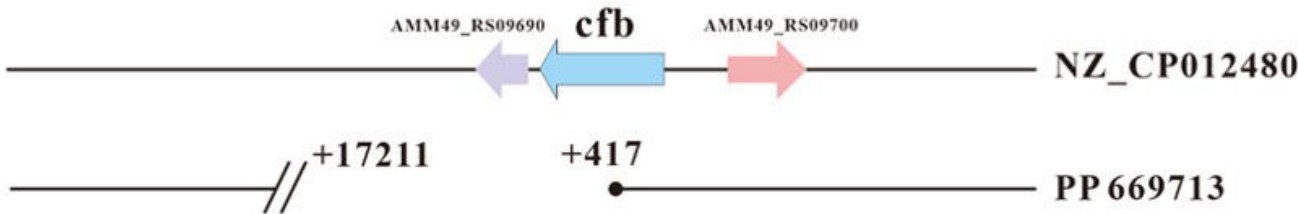

**FIG 3** Whole-genome sequencing analysis of strain PP669713. The C-terminus of the *cfb* gene is missing, with the bottom box indicating the retained sequence of the *cfb* gene.

by incorporating tailored nutrients and indicators that stimulate its growth. The resultant color change in these media facilitates swift identification of GBS colonies. However, this approach hinges on the β-hemolysin production capacity of GBS, a trait absent in certain strains. The second type is Media with Specific Enzyme Substrates. This medium consists of a basic agar enriched with precise enzyme substrates. Upon metabolism by GBS, these substrates release chromogens, inducing a color change in the colonies. This method boasts significant advantages in sensitivity and accuracy, as it can detect both β-hemolytic and non-β-hemolytic GBS strains. Research indicates that media relying solely on β-hemolysis characteristics may miss detecting 1%–4% of GBS isolates (39, 40). A study conducted by the U.S. Centers for Disease Control and Prevention (CDC) revealed that, between 2006 and 2008, of the 265 GBS isolates sourced from early-onset infections, 11 (approximately 4.1%) were non-hemolytic (41). These findings underscore the significance of selecting an appropriate culture medium to guarantee precise detection and identification of GBS. Given the limitations of β-hemolysis-based media, particularly

**TABLE 2** Comparison of antibiotic resistance between CAMP-positive and CAMP-negative GBS isolates[a]

| Antibiotic | CAMP-positive (n = 490) | CAMP-negative (n = 19) | Fisher's exact test (P-value) | Bootstrap (P-value) | Relative risk (95% CI) | Bayesian posterior probability |
|---|---|---|---|---|---|---|
| Penicillin | 475/490 (96.9%) | 19/19 (100%) | 1.000 | 1.000 | 1.00 (1.00–1.00) | 0.50 |
| Ampicillin | 488/490 (99.6%) | 19/19 (100%) | 1.000 | 1.000 | 1.00 (1.00–1.00) | 0.50 |
| Linezolid | 486/490 (99.2%) | 19/19 (100%) | 1.000 | 1.000 | 1.00 (1.00–1.00) | 0.50 |
| Quinupristin | 490/490 (100%) | 18/19 (94.7%) | 0.000 | 0.000 | 0.00 (0.00–0.00) | 0.99 |
| Ciprofloxacin | 166/490 (33.9%) | 0/19 (0%) | 0.002 | 0.002 | 0.00 (0.00–0.00) | 0.99 |
| Levofloxacin | 161/490 (32.9%) | 0/19 (0%) | 0.003 | 0.003 | 0.00 (0.00–0.00) | 0.99 |
| Clindamycin | 345/489 (70.6%) | 13/19 (68.4%) | 0.842 | 0.839 | 0.97 (0.76–1.24) | 0.52 |
| Tetracycline | 399/490 (81.4%) | 12/19 (63.2%) | 0.048 | 0.056 | 1.27 (0.99–1.63) | 0.63 |

[a]P-values were calculated using Fisher's exact test for small sample sizes and Bootstrap resampling for robust estimation. A P-value < 0.05 is considered statistically significant. Relative risk (RR): the ratio of the probability of antibiotic resistance in CAMP-positive isolates to that in CAMP-negative isolates, with 95% confidence intervals (CI). An RR > 1 indicates a higher risk of resistance in CAMP-positive isolates. Bayesian posterior probability: the probability that the resistance rate in CAMP-positive isolates is higher than in CAMP-negative isolates, based on Bayesian analysis. Sample sizes: the number of isolates tested for each antibiotic is provided in parentheses (e.g., n = 490 for CAMP-positive and n = 19 for CAMP-negative). Missing data: some antibiotics had incomplete data for CAMP-positive isolates due to technical limitations or missing tests. All CAMP-negative isolates were fully tested.

their inability to detect non-β-hemolytic GBS strains, the employment of chromogenic media with specific enzyme substrates is more advantageous. These media offer a more comprehensive and reliable approach for detecting both hemolytic and non-hemolytic GBS, thereby enhancing the accuracy of GBS screening and diagnosis.

Notably, our epidemiological data revealed a significantly greater proportion of pregnant women with overweight BMI status among those with CAMP-negative GBS strains than among those with CAMP-positive strains. While this finding suggests a potential link between maternal characteristics and the CAMP phenotype, it is important to interpret these results with caution. The observed association could be due to chance, given the relatively small sample size and the specific population studied. Further research, including larger and more diverse cohorts, is needed to determine whether this association is consistent and biologically meaningful. Given the increasing isolation of CAMP-negative GBS strains, it is essential to explore the factors contributing to the presence of these *cfb*-deficient strains in future research.

Importantly, we used two qPCR assay kits: one targeting a single gene (*cfb*) and the other targeting multiple genes (*cfb* and *cps*). As anticipated, all the isolates, except for one strain, tested positive with the multitarget qPCR kit but negative with the single-target kit. Our findings highlight the necessity of targeting multiple conserved chromosomal regions of GBS for accurate molecular diagnosis. The inclusion of additional targets, such as cps, can mitigate the risk of false negatives, thereby improving the reliability of GBS detection. Given that some GBS strains are camp-negative and do not produce the CAMP factor, single-target assays that only focus on the *cfb* gene may fail to identify these strains, potentially leading to underdiagnosis. To enhance the clinical relevance of our findings, it is imperative to review the package inserts of commercially available GBS molecular screening tests. Such an evaluation would reveal which tests rely on a single target (cfb) and which employ a dual-target approach. This information is vital for identifying potential sources of false-negative results in current diagnostic practices and for guiding improvements in testing methodologies to ensure more accurate and comprehensive GBS detection.

All 19 CAMP-negative GBS isolates in our study presented deficiency of the *cfb* gene. This finding aligns with the results reported by Zhou et al. but contrasts with those of Guo et al. (27, 33). In Guo's research, five CAMP-negative GBS strains were identified, four of which tested positive for the *cfb* gene via PCR. This finding suggests that the negative CAMP test for these four strains may stem from inefficient expression of the CAMP factor. However, Guo's study did not utilize whole-genome sequencing (WGS) or additional PCR assays on other regions of the *cfb* gene. Therefore, it remains unclear whether these four strains retain part of the *cfb* gene, which could also explain the positive PCR results.

Interestingly, unlike previous reports of chromosomal deletions in CAMP-negative GBS strains (31, 32), 18 of our isolates exhibited a complete loss of the *cfb* gene. One isolate (strain PP669713) retained the upstream region but lost the entire downstream portion. This partial deletion led to the strain's negative CAMP reaction but still produced positive qPCR results for the upstream region. This is the first report of a C-terminal deletion of the *cfb* gene in a CAMP-negative GBS strain, confirming that both the N-terminal and C-terminal regions of the CAMP factor are essential for its cohemolytic activity (42). Further investigation using advanced techniques such as WGS would be necessary to elucidate the precise genetic basis of this phenotype.

According to multilocus sequence typing (MLST) analysis, all CAMP-negative GBS strains in our study were serotype III and belonged to the ST862 sequence type, with the exception of one isolate for which the sequence type could not be determined. Notably, ST862 has previously been identified in the same city by Liang et al., who reported it as the second most prevalent sequence type in the Fuzhou region (43). This finding suggests that ST862 may represent a locally dominant clonal lineage within this geographic area. As illustrated in Table S1, nearly all ST862 strains isolated in China are concentrated in the southern region, indicating a potential association between GBS sequence types and geographical distribution. Furthermore, Table S1 reveals that almost

all ST862 strains isolated across Asia are serotype III, with only one strain identified as serotype Ib. Remarkably, all ST862 isolates to date have originated from Asia, with the vast majority sourced from China. These observations highlight the regional specificity of ST862 and its strong association with serotype III, underscoring the importance of considering both geographical and serotypic factors when studying GBS epidemiology.

Consistent with prior studies, all GBS isolates in our research were found to be sensitive to penicillin, ampicillin, and linezolid. The data reveal that CAMP-positive isolates exhibit significantly higher resistance rates to quinupristin, ciprofloxacin, and levofloxacin compared to CAMP-negative isolates. This suggests that these antibiotics may not be optimal choices for treating CAMP-positive GBS infections. Additionally, a marginally significant increase in tetracycline resistance was observed in CAMP-positive isolates, which warrants further investigation to confirm this trend. These findings underscore the critical importance of prudent antibiotic selection in clinical practice and highlight the need for continuous surveillance and research to refine management strategies for GBS infections. Understanding the molecular mechanisms underlying these resistance patterns and exploring regional variations will be essential for developing evidence-based treatment guidelines.

While the findings provide valuable insights into the antibiotic resistance profiles of CAMP-positive and CAMP-negative GBS isolates, it is important to acknowledge the limitations of the study. The substantial discrepancy in sample sizes between CAMP-positive isolates ($n = 490$) and CAMP-negative isolates ($n = 19$) could potentially have influenced the statistical robustness of the analysis. Although we utilized Bootstrap Resampling to address this concern, future studies incorporating larger sample sizes of CAMP-negative isolates would yield more definitive results.

Furthermore, future research should delve into the molecular mechanisms underlying these resistance patterns and investigate regional variations to inform the development of evidence-based treatment guidelines. Understanding the molecular basis of increased resistance in CAMP-positive isolates could lead to the development of new therapeutic strategies or diagnostic tools, while exploring regional variations will help tailor treatment protocols to specific geographic areas.

There are several limitations in our study. First, we determined that the 19 isolated phenotypically CAMP-negative GBS strains resulted from chromosomal deletion of the *cfb* gene. However, we did not elucidate the specific sizes and sites of these deletions in all the strains. To address this, whole-genome sequencing should be conducted on the remaining 18 strains in future studies.

Second, whereas previous research has indicated that CAMP is not essential for the virulence and colonization of GBS (44, 45), clinical data on the effects of CAMP-negative GBS strains remain limited. The pathogenicity of *cfb* gene-deficient strains, particularly strain PP669713, which retains the N-terminal portion of the *cfb* gene, requires further investigation. Future studies should focus on two key questions regarding this N-terminal retaining strain: (i) can the incomplete *cfb* gene still express a functional protein and (ii) if so, what role this encoded protein plays? Additionally, the significant disparity between the number of CAMP-positive and CAMP-negative strains may bias direct comparisons between these groups. This limitation could be mitigated in future research by the isolation of more CAMP-negative GBS strains.

Overall, whereas the frequency of mutations and chromosomal deletions in the *cfb* gene varies across geographical regions—ranging from less than 1% to 7.9%—it is essential to continue monitoring the molecular characteristics of evolving GBS strains through ongoing clinical surveillance. Such efforts will increase the accuracy and reduce the failure rate of molecular diagnostic methods (32, 33).

## Conclusion

Our study isolated 19 CAMP-negative strains from a total of 526 GBS strains and revealed that the CAMP-negative phenotype resulted from either complete or partial deletion of the CAMP-encoding gene *cfb*. Further research is warranted to determine whether the

partially retained *cfb* gene can still produce a functional protein and, if so, the role of this protein. Furthermore, the 3.6% isolation rate of CAMP-negative GBS suggests that these strains may be more widespread than previously acknowledged. To improve diagnostic accuracy, a multitarget PCR assay that focuses on at least two conserved genes within the GBS genome would be an optimal strategy for future molecular GBS testing.

## ACKNOWLEDGMENTS

We would like to thank all the individuals who participated in the study and all the staff members and physicians at Fujian Provincial Hospital South Branch who assisted with implementing this project.

This study was supported by grants from the Startup Fund for Scientific Research of Fujian Medical University (No. 2022QH1327 to L.Z. and No. 2021QH1274 to M.C.) and the Fujian Provincial Health Technology Project of Fujian Provincial Health Commission (No. 2023QNA005 to M.C.).

L.Z. and X.L. designed the experiments. L.Z., X.L., and M.C. executed the experiments. L.Z., X.L., M.C., J.W., and H.L. performed the data analysis. J.W., H.X., W.H., and D.C. collected the clinical samples and information. L.Z. and X.L. wrote the manuscript with input from all of the other authors. All the authors contributed to the article and approved the submitted version.

## AUTHOR AFFILIATIONS

[1]Department of Clinical Laboratory, Fujian Provincial Hospital, Fuzhou University Affiliated Provincial Hospital, Fuzhou, China
[2]Shengli Clinical Medical College, Fujian Medical University, Fuzhou, China
[3]Central Laboratory, Fujian Provincial Hospital, Center for Experimental Research in Clinical Medicine, Fuzhou, China
[4]Fujian Provincial Key Laboratory of Critical Care Medicine, Fujian Provincial Key Laboratory of Cardiovascular Disease, Fuzhou, China

## AUTHOR ORCIDs

Xixi Lai http://orcid.org/0009-0007-9265-2194
Pengwei Cai http://orcid.org/0009-0007-6935-149X
Lilan Zheng http://orcid.org/0009-0001-0325-3930

## FUNDING

| Funder | Grant(s) | Author(s) |
| --- | --- | --- |
| Fujian Provincial Health Technology Project | NO.2023QNA005 | Meihong Chen |
| Fujian Medical University | NO.2021QH1274 | Meihong Chen |
| Fujian Medical University | NO.2022QH1327 | Lilan Zheng |

## AUTHOR CONTRIBUTIONS

Xixi Lai, Conceptualization, Data curation, Formal analysis, Investigation, Methodology, Resources, Validation, Visualization, Writing – original draft, Writing – review and editing | Meihong Chen, Conceptualization, Data curation, Funding acquisition, Investigation, Methodology, Resources, Validation, Visualization, Writing – review and editing | Jianwei Wang, Formal analysis, Software | Junjun Wang, Data curation, Methodology | Hui Lv, Formal analysis, Software | Haihua Xie, Methodology | Wenjuan He, Data curation,

Investigation | Dongjie Chen, Methodology | Yi Huang, Writing – review and editing | Pengwei Cai, Writing – review and editing | Lilan Zheng, Data curation, Funding acquisition, Investigation, Methodology, Validation, Visualization, Writing – review and editing

## DATA AVAILABILITY

The data sets used in this study can be found in NCBI under accession no. PP669702, PP669703, PP669704, PP669705, PP669706, PP783523, PP669707, PP669708, PP669709, PP669710, PP669711, PP669712, PP669713, PP669714, PP669716, PP783524, PP669715, PP783526, and PP783527.

## ETHICS APPROVAL

This study was approved by the Ethics Review Committee of Fujian Provincial Hospital. Informed written consent was not obtained, as the samples were collected during the regular GBS screening of pregnant women who were not at risk. No patients were directly involved in the study process or asked questions in the study.

## ADDITIONAL FILES

The following material is available online.

### Supplemental Material

**Table S1 (Spectrum03257-24-s0001.docx).** The geographic distribution of current isolated ST862 GBS.

### Open Peer Review

**PEER REVIEW HISTORY (review-history.pdf).** An accounting of the reviewer comments and feedback.

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
