## [Reviewer comments · Microbiology Spectrum]

Microbiology Spectrum

CAMP-negative *Streptococcus agalactiae* strains exhibited complete or partial chromosomal deletions of the CAMP-factor encoding gene *cfb*

Xixi Lai, Meihong Chen, Jianwei Wang, Junjun Wang, Hui Lv, Haihua Xie, Wenjuan He, Dongjie Chen, Yi Huang, Pengwei Cai, and Lilan Zheng

Corresponding Author(s): Xixi Lai, Fuzhou University Affiliated Provincial Hospital

Review Timeline:

Submission Date:	December 26, 2024
Editorial Decision:	February 6, 2025
Revision Received:	February 13, 2025
Accepted:	March 10, 2025

Editor: Cecilia Thompson

Reviewer(s): The reviewers have opted to remain anonymous.

Transaction Report:

DOI: <https://doi.org/10.1128/spectrum.03257-24>

Re: Spectrum03257-24 (CAMP-negative *Streptococcus agalactiae* strains exhibited complete or partial chromosomal deletions of the CAMP-factor encoding gene *cfb*)

Dear Ms. Xixi Lai:

Thank you for the privilege of reviewing your work. Below you will find my comments, instructions from the Spectrum editorial office, and the reviewer comments.

The revised manuscript is well written however, there are a few minor comments below that should be addressed:

Add space between percentage and following word in lines 35-36: From 5794 samples, 526 (9.1%) GBS strains, including 19 (3.6%) CAMP-negative strains and two strains from the same patient, were isolated.

Remove lines 86-91 as they are redundant with the previous paragraph. Recommend amending line 78 to state: This approach, considered universal screening, aims to identify and treat all GBS-colonized women with antibiotics during labor to prevent early-onset GBS disease in their infants.

Move lines 313-321 to the discussion. Recommend incorporating into paragraph at lines 426-441.

Switch paragraphs (lines 363-370 and lines 371-384) in the discussion so that the section starts with the discussion of the colonization rate followed by the use of the CAMP test to identify isolates.

Add space between 'strains.' and 'The' in lines 391.

Add space between 'strains.' and 'Research' in lines 396.

Recommend using 'more advantageous' instead of 'advisable' in line 404.

Recommend altering text to 'overweight BMI status' in line 408.

Add space between 'studied.' and 'Further' in line 413.

Add to figure legend 1A and 1B to briefly describe image. For example in 1A: Production of CAMP by GBS causes arrow-shaped hemolysis in the presence of *Staphylococcus aureus* on sheep blood agar. For example in 1B: GBS metabolizes chromogenic substances resulting purple pigmented colonies on the chromogenic agar.

Revision Guidelines

Data availability: ASM policy requires that data be available to the public upon online posting of the article, so please verify all links to sequence records, if present, and make sure that each number retrieves the full record of the data. If a new accession number is not linked or a link is broken, provide Spectrum production staff with the correct URL for the record. If the accession numbers for new data are not publicly accessible before the expected online posting of the article, publication may be delayed;

please contact production staff (Spectrum@asmusa.org) immediately with the expected release date.

Sincerely,
Cecilia Thompson
Editor
Microbiology Spectrum

Response to Reviewers

1. Regarding Lines 35-36

Reviewer Comment: Add a space between percentages and the following word.

Response: We have added spaces between percentages and the following words as suggested by the reviewer to improve readability.

2. Deleting Lines 86-91

Reviewer Comment: Delete lines 86-91 as they are redundant with the previous paragraph.

Response: We have removed lines 86-91 to eliminate redundancy and ensure the text remains concise and clear.

3. Revising Line 78

Reviewer Comment: Amend line 78 to state: "This approach, considered universal screening, aims to identify and treat all GBS-colonized women with antibiotics during labor to prevent early-onset GBS disease in their infants."

Response: We have revised line 78 as suggested by the reviewer to clarify the purpose of the universal screening approach.

4. Moving Lines 313-321 to the Discussion

Reviewer Comment: Move lines 313-321 to the discussion section and incorporate them into the paragraph at lines 426-441.

Response: We have relocated lines 313-321 to the discussion section and integrated them into the paragraph at lines 426-441 to enhance logical coherence.

5. Rearranging Paragraphs in the Discussion

Reviewer Comment: Switch the order of paragraphs (lines 363-370 and lines 371-384) in the discussion so that the section starts with the colonization rate followed by the use of the CAMP test to identify isolates.

Response: We have rearranged the paragraphs in the discussion section as suggested to ensure a more logical flow of information.

6. Adding Spaces in Lines 391 and 396

Reviewer Comment:

Add a space between "strains." and "The" in line 391.

Add a space between "strains." and "Research" in line 396.

Response: We have added spaces in the specified locations to improve sentence fluency.

7. Adjusting Word Choice in Line 404

Reviewer Comment: Replace "advisable" with "more advantageous" in line 404.

Response: We have replaced "advisable" with "more advantageous" as suggested to better convey the intended meaning.

8. Adjusting Terminology in Line 408

Reviewer Comment: Alter the text to "overweight BMI status" in line 408.

Response: We have replaced "overweight" with "overweight BMI status" to provide a more precise description of the target population.

9. Adding Space in Line 413

Reviewer Comment: Add a space between "studied." and "Further" in line 413.

Response: We have added a space in the specified location to optimize sentence structure.

10. Adding Descriptions to Figure Legends 1A and 1B

Reviewer Comment: Add brief descriptions to figure legends 1A and 1B. For example:

Figure 1A: Production of CAMP by GBS causes arrow-shaped hemolysis in the presence of *Staphylococcus aureus* on sheep blood agar.

Figure 1B: GBS metabolizes chromogenic substances resulting in purple pigmented colonies on the chromogenic agar.

Response: We have added brief descriptions to figure legends 1A and 1B as suggested to help readers better understand the content of the images.

Conclusion

We have carefully addressed all of the reviewer's comments and made the necessary adjustments or additions to the respective sections. These revisions not only improve the quality and readability of the manuscript but also enhance its logic and professionalism. We sincerely thank the reviewer for their valuable feedback!

Re: Spectrum03257-24R1 (**CAMP-negative *Streptococcus agalactiae* strains exhibited complete or partial chromosomal deletions of the CAMP-factor encoding gene *cfb***)

Dear Ms. Xixi Lai:

Your manuscript has been accepted, and I am forwarding it to the ASM production staff for publication. Your paper will first be checked to make sure all elements meet the technical requirements. ASM staff will contact you if anything needs to be revised before copyediting and production can begin. Otherwise, you will be notified when your proofs are ready to be viewed.

Sincerely,
Cecilia Thompson
Editor
Microbiology Spectrum